# Bypassing ubiquitination enables LAT recycling to the cell surface and enhanced signaling in T cells

**Lakshmi Balagopalan**  *, **Hiba Malik, Katherine M. McIntire, Joseph A. Garvey, Tiffany Nguyen, Ana B. Rodriguez-Peña, Lawrence E. Samelson** *

Laboratory of Cellular and Molecular Biology, Center for Cancer Research, National Cancer Institute, National Institutes of Health, Bethesda, Maryland, United States of America

* balagopl@mail.nih.gov (LB); samelsonl@helix.nih.gov (LES)

## Abstract

LAT molecules defective in ubiquitination have an increased half-life and induce enhanced signaling when expressed in T cells. In this study, we have examined the role of ubiquitination in regulating LAT endocytosis, recycling, and degradation in resting and stimulated T cells. By tracking and comparing plasma membrane-labeled wild type and ubiquitination-resistant 2KR LAT, we find that ubiquitination promotes the degradation of surface LAT in T cells. Activation of T cells increases LAT ubiquitination and promotes trafficking of internalized LAT to lysosomes for degradation. Ubiquitination of LAT does not change internalization rates from the cell surface, but prevents efficient recycling of LAT to the surface of T cells. Our study demonstrates that surface LAT levels are tightly controlled by ubiquitination. LAT in unstimulated cells lacks ubiquitin allowing for increased LAT stability and efficient T cell activation upon TCR triggering; ubiquitination leads to efficient removal of LAT after activation.

**Data Availability Statement:** All relevant data are within the paper and its Supporting Information files.

## Introduction

T cell activation is mediated by engagement of the T Cell antigen Receptor (TCR). Phosphorylation of the TCR complex on cytosolic tyrosine residues leads to the binding and activation of a Syk-family protein tyrosine kinase (PTK), zeta-chain-associated protein kinase 70 (ZAP-70), which in turn phosphorylates key adapter proteins including the transmembrane protein, Linker for Activation of T cells (LAT) [1]. The LAT cytoplasmic domain contains several conserved tyrosine residues, which upon phosphorylation by ZAP-70, provide docking sites for the recruitment of other adapters (e.g., Grb2, SLP-76), enzymes (e.g., PLCg1, Vav), and the regulatory subunit of PI3K, resulting in the assembly of multiprotein complexes. These signaling complexes transduce and propagate TCR signals, leading to activation of the downstream effectors that mediate outcomes such as T cell proliferation and cytokine expression [2]. A hallmark of T cell activation is the rapid formation of microclusters that act as platforms for the recruitment and activation of downstream effector molecules. Microclusters are enriched in phosphorylated signaling proteins, and function as basic signaling units for T cell activation [3]. Soon after recruitment to microclusters, signaling molecules including LAT and SLP-76

**Funding:** This research was supported by the Intramural Research Program of the NIH, NCI, CCR.

**Competing interests:** The authors have declared that no competing interests exist.

from microclusters are rapidly internalized in a process dependent on the E3 ligase c-Cbl and ubiquitin [4, 5], thus tightly regulating T cell signaling.

Studies showed that, in addition to phosphorylation, the LAT cytoplasmic tail is also subject to ubiquitination upon T cell stimulation [4, 6, 7]. To elucidate the biological role of LAT ubiquitination, we substituted LAT lysines with arginines to generate 2KR LAT. Expression of this mutant LAT resulted in a dramatic decrease in overall LAT ubiquitination, and ubiquitination-resistant 2KR LAT mutants displayed a decrease in protein turnover rates [8]. Importantly, T-cell signaling was elevated in cells expressing this LAT mutant and in T cells from transgenic mice expressing these mutants, indicating that inhibition of LAT ubiquitylation in T cell lines and primary T cells enhances T-cell signaling [8–10]. These results support LAT ubiquitylation as a molecular checkpoint for attenuation of T-cell signaling.

An increasingly important concept for understanding LAT function is membrane trafficking. LAT is localized at the plasma membrane and also in intracellular vesicles in resting and stimulated cells [11, 12]. The relative importance of plasma membrane-localized LAT versus vesicular LAT for microcluster formation and TCR activation has been extensively studied. In one model, direct recruitment of cell surface LAT to microclusters is critical for T-cell activation [13–15], while in another model, vesicular, but not cell surface LAT, is essential [16–19]. Recently, we used lattice light sheet microscopy to image the sequence of events in microcluster formation. We observed that cell surface LAT is rapidly recruited into microclusters and phosphorylated at sites of T-cell activation, and that the vesicular pool is recruited subsequently [20]. Retrograde traffic of LAT from the cell surface to the Golgi is also important for LAT delivery to the immune synapse and T cell activation [21]. Thus, phosphorylation of LAT present at the plasma membrane triggers various downstream signaling cascades and the amount of cell surface LAT could determine the magnitude of T cell activation.

In this study, we investigated the relationship between LAT ubiquitination, LAT endocytic trafficking, and surface LAT expression in T cells. We found no correlation between the capacity for LAT ubiquitination and the overall rate of LAT endocytosis. However, ubiquitination prevented the efficient recycling of internalized LAT back to the plasma membrane. Furthermore, we found that ubiquitination regulated LAT levels by promoting the degradation of internalized LAT in lysosomes. Our data demonstrate that ubiquitination diverts recycling LAT to a pathway leading to lysosomal degradation in activated T cells, thereby providing a mechanism for the selective turnover of LAT complexes.

## Materials and methods

### Reagents

Human anti-CD3ε antibodies (BD Pharmingen) were used to coat coverslips. The following antibodies were used: rabbit anti-CD4 (Sigma), mouse anti-LAT (Upstate), rabbit anti-pLAT[191] (Invitrogen), mouse anti- PLCγ1, mouse SLP-76 (Antibody Solutions), mouse anti-pY 4G10 (Millipore). Alexa-488 or APC-conjugated anti-CD4 [clones OKT4 and SK3 (eBioscience)] were used for labeling CD4. Indo-1 AM was from Invitrogen. TAK-243 to inhibit E1 ubiquitin activating enzyme was from Selleckchem.

### Cell culture and generation of stable cell lines

Jurkat E6.1 and LAT-deficient JCam2.5 cells have been described previously [22]. For stable cell lines, Jurkat cells were retrovirally transduced with CD4-WTLAT or CD4-2KRLAT constructs cloned in a retroviral plasmid pMSCVneo (CD4-WTLAT has been previously described in [13]. Following transduction, cells were cultured in medium containing 1.8mg/ml G418 for 2 weeks, and were then sorted for CD4 expression.

## Immunoprecipitation and immunoblotting

For immunoprecipitations, stable cell lines were lysed in ice-cold lysis buffer containing 1% Brij, 1% n-octyl-D-glucoside, 50 mM Tris HCl, pH 7.6, 150 mM NaCl, 5 mM EDTA, 1 mM Na3VO4 and complete protease inhibitor tablets (Roche). Cellular lysates were subjected to immuno- precipitation using an anti-LAT rabbit antibody. Protein A/G Plus agarose beads (Santa Cruz Biotechnology) were used for immunoprecipitation. A total of 90% of the protein samples were resolved by SDS-polyacrylamide gel electrophoresis (SDS/ PAGE), transferred to nitrocellulose membrane, and analyzed by Western blot for total LAT. To analyze the phos- phorylation status of various signaling proteins, E6.1 Jurkat cells or JCam2.5 cell lines stably expressing CD4-WTLAT and CD4-2KRLAT were left untreated or treated with anti-CD3 (OKT3 ascites, 1 μg/mL) for various time points and lysed using a twofold excess of boiling 2× sample buffer [20 mM Tris (pH 8.0), 2 mM EDTA, 2 mM Na3- VO4, 20 mM DTT, 2% SDS, and 20% glycerol]. The samples were then heated at 95 ˚C for 5 min and sonicated to reduce the viscosity of the solution. $7 \times 10^5$ cell equivalents were separated by PAGE using 10% Crite- rion Precast polyacrylamide gels. The separated proteins were then transferred to polyvinyli- dene di- fluoride and the membrane was blocked for 1 h at room temperature using TBST [10 mM Tris (pH 8.0), 150 mM NaCl, and 0.05% Tween 20] with 5% milk and 1% BSA. The mem- branes were incubated overnight at 4 ˚C with primary Abs diluted in TBST with 5% milk, 1% BSA, followed by a 60-min incubation at room temperature with the appropriate secondary Ab diluted in TBST with 5% milk, 1% BSA. The blots were then visualized by chemiluminescence.

## Confocal microscopy and image processing

For antibody labeling of surface LAT, cells were re-suspended in cold buffer (RPMI + 0.5% BSA + 25mM Hepes) and incubated on ice for 10 minutes. Surface CD4-LAT was stained with CD4-APC antibody (clone OKT4) on ice for 30 min. After three washes in cold buffer to remove excess antibody, labeled cells were resuspended into warm imaging buffer (RPMI without phenol-red + 10%FBS + 25mM Hepes) and dropped immediately onto anti-CD3ε coverslips. Spreading assays were performed as described previously (Bunnell et al., 2002). Images from fixed cells were collected with a Zeiss 780 LSCM, using a 63X, 1.4 NA objective (Carl Zeiss Inc.). Imaris 7.2.3 (Bitplane, Andor) was used for initial image processing and to calculate Pearson's Correlation Coefficient (PCC) values and internal versus surface ratios. Adobe PhotoShop and Illustrator (Adobe Systems Inc.) were used to prepare composite figures.

## Calcium flux and CD69 assays

For measurement of calcium influx, cells were incubated with 5uM Indo-I AM (Molecular probes) and 0.5mM probenecid (Sigma) in RPMI medium 1640 at 37 ˚C for 45 min. The cells were washed with RPMI 1640, resuspended in imaging buffer containing 0.5 mM probenecid and 20 mM Hepes in RPMI 1640 without phenol red. The cells were incubated at 37 ˚C for 5 min before measurements, stimulated with 10ug of soluble OKT3 antibody and analyzed using the LSR II (BD Biosciences). For measurement of surface CD69 levels in Jurkat cells, 1x10.6 Jurkat cells were stimulated with 10ug/ml plate-bound OKT3. Sixteen hours post-stimulation, cells were stained with APC-conjugated anti-CD69 and surface expression was analyzed on a FACSCaliber (BD Biosciences). The data were processed using Tree Star FlowJo software and graphed using Excel.

## Surface loss assay

JCam2.5 cells stably expressing CD4-WTLAT or CD4-2KRLAT were surface-labeled with CD4-APC as follows:1.5X10.6 cells/tube were spun down for each time point of the internalization assay (0, 60 and 120 minutes stimulated and un-stimulated time points). Cells were re-suspended in cold buffer (RPMI+ 0.5% BSA + 25mM Hepes) and incubated on ice for 10 minutes. Surface CD4-LAT was stained with CD4-APC antibody (clone OKT4) on ice for 30 min. Excess antibody was washed off three times in cold RPMI. After removing a sample for time 0 measurement ($T_0$), cells were re-suspended in warm buffer and transferred to 37˚C water bath. At various time points ($T_x$), cells were removed and transferred to cold RPMI to stop internalization. Following completion of the time course all samples were fixed in 4% PFA for 20 minutes on ice. Cells were washed with cold RPMI, resuspended in FACS buffer and analyzed by flow cytometry on a BD FACS Caliber. Surface loss was measured as ($T_0$- $T_x$)/ $T_0$ X 100. $T_0$ represents the mean fluorescence intensity of cells before transfer to 37˚C and $T_x$ is the mean fluorescence intensity of cells at each time point.

## Endocytosis assay

JCam2.5 cells stably expressing CD4-WTLAT or CD4-2KRLAT were surface-labeled with CD4-APC as follows:1.5 million cells/tube were spun down for each time point of the internalization assay (0, 5, 10 and 15 minutes stimulated and un-stimulated time points). Cells were re-suspended in cold internalization buffer (RPMI+ 0.5% BSA + 25mM Hepes) and incubated on ice for 10 minutes. Surface CD4-LAT was stained with CD4-APC antibody (clone OKT4) on ice for 30 min. Excess antibody was washed off three times in cold RPMI and cells were re-suspended in cold internalization buffer. Two samples were removed for $T_0$ and $T_0$ stripped control. Remaining cells were re-suspended in warm 37˚C internalization buffer and immediately put in 37˚C water bath. Cells were removed at each time point and transferred to cold RPMI on ice to stop internalization. Following completion of the time course, all samples (except for $T_0$) were re-suspended in stripping buffer (pH 2.0 0.5% BSA in 1X PBS) for 1.5 minutes and washed with cold RPMI. All samples were fixed in 4% PFA for 20 minutes on ice. Cells were washed with cold RPMI and resuspended in FACS buffer and analyzed by flow cytometry on a BD FACS Caliber. Endocytosis was measured as ($T_0$- $T_x$)/ $T_0$ X 100. $T_0$ represents the mean fluorescence intensity of unstripped cells and $T_x$ is the mean fluorescence intensity of cells stripped at each time point.

## Recycling assay

JCam2.5 cells stably expressing CD4-WTLAT or CD4-2KRLAT were surface-labeled with CD4-APC antibody as described above. Excess antibody was washed off three times in cold RPMI and cells were resuspended in cold recycling buffer (RPMI+ 0.5% BSA and 25mM Hepes). Cells were re-suspended in warm 37˚C recycling buffer and immediately put in 37˚C water bath for 30 minutes to allow internalization of labeled LAT. After 30 minutes, cells were spun down at 4˚C, and pellets re-suspended in cold stripping buffer (pH 2.0 0.5% BSA in 1X PBS) for 1.5 min to remove antibody on the cell surface. Cells were washed with excess cold RPMI and re-suspended in cold recycling buffer and kept on ice. Cells were removed from each WT LAT and 2KR LAT sample for 0 time-point control and put in cold RPMI to stop the recycling process. The remaining volume of WT LAT and 2KR LAT cells were divided into two halves. One half was resuspended in warm recycling buffer with stimulatory OKT3 (10μg/ ml), and the other half in warm recycling buffer without OKT3. Tubes were returned to the 37˚C water bath. Resuspended cells were then incubated for various times (5,10 and 15 minutes) to allow for recycling of internalized LAT. Cells were removed at each time point and

transferred to cold RPMI on ice. Following completion of the time course all samples were re-suspended in stripping buffer for 1.5 minutes and washed with cold RPMI. Next, all pellets were resuspended and fixed in 4% PFA for 20 minutes on ice. Cells were washed with cold RPMI and resuspended in FACS buffer and analyzed by flow cytometry on a BD FACS Caliber. The percentage of recycled LAT was measured using the equation $1-(T_0- T_x)/ T_0 \times 100$. $T_0$ represents the mean fluorescence intensity of cells following the second acid trip at time zero and $T_x$ is the mean fluorescence intensity of cells stripped at each time point.

## Results and discussion

### CD4-2KRLAT displays enhanced signaling and decreased degradation from the cell surface

To specifically follow LAT trafficking from the cell surface, we used a CD4-WTLAT chimera [13] in which the extracellular domain of CD4 was fused to the transmembrane and cytosolic regions of LAT (Fig 1A, top). For the current study, we substituted the cytosolic lysines of LAT with arginines to generate CD4-2KRLAT (Fig 1A, bottom). We generated stable cell lines expressing CD4-WTLAT and CD4-2KRLAT in LAT-deficient JCam2.5 Jurkat cells. Using an antibody to CD4 in intact cells, we detected increased levels of CD42KRLAT compared to CD4WTLAT on the cell surface of our stable cell lines (Fig 1B, left). For this study. we sorted to get equivalent surface levels of CD4 proteins (Fig 1B, right). Both the CD4-WTLAT and CD4-2KRLAT chimeras retained the ability to interact with the signaling proteins SLP-76 and PLC-γ1 upon T cell activation (S1A Fig). Moreover, both CD4-WTLAT and CD4-2KRLAT were recruited to phosphotyrosine-containing microclusters (S1B Fig). When sorted populations were evaluated for TCR signaling, we observed enhanced TCR-induced cytosolic calcium flux in LAT-deficient JCam2.5 cells expressing CD4-2KRLAT compared to cells expressing CD4-WTLAT (Fig 1C). These results indicate that this molecule retains enhanced LAT functions that were demonstrated for ubiquitin-resistant LAT molecules in cell lines, transgenic mice and primary human T cells previously (Fig 1C and 1D, [8–10]).

The increased cell surface levels of CD4-2KRLAT could be a consequence of the increased lifetime of 2KRLAT molecules at the plasma membrane, in vesicles or both. To specifically follow the lifetime of LAT molecules at the cell surface, we tracked surface-labeled CD4-WTLAT or CD4-2KRLAT using fluorescent anti-CD4. We used clone OKT4 to follow CD4-LAT molecules as we have previously shown that it does not cause activation of T cells [13]. Cell surface proteins on T cells were labeled using fluorescently-labelled anti-CD4 on ice, the cells were washed, and the T cells were retained on ice or transferred to 37˚C. At various times, the cellular anti-CD4 fluorescent signal was measured using flow cytometry. The level of fluorescence revealed the total amount of anti-CD4 (i.e., cell surface and intracellular) present in the cell and thus indirectly assesses LAT degradation or LAT incorporation into an endosomal compartment where the fluorescence is quenched. Using this assay, we found that the signal from the WTLAT initially at the cell surface decreased to about 78% of starting levels after 2 hrs of incubation at 37˚C (Fig 1D). 2KR LAT molecules, which are resistant to ubiquitination, were lost at a significantly lower rate than WT LAT, retaining 90% of initial surface levels after 2hrs. As it has been reported that LAT is ubiquitinated upon T cell stimulation [6, 7], we evaluated the effect of TCR stimulation on the loss of surface WT and 2KR LAT. TCR stimulation significantly increased the loss of surface labeled WTLAT, without significantly affecting the loss of 2KR LAT at 60 minutes (Fig 1D). At 120 minutes, the difference in loss between unstimulated and stimulated WT LAT was even more apparent, while the difference in loss of 2KR LAT in unstimulated and stimulated samples was not as remarkable. The stimulation-dependent loss

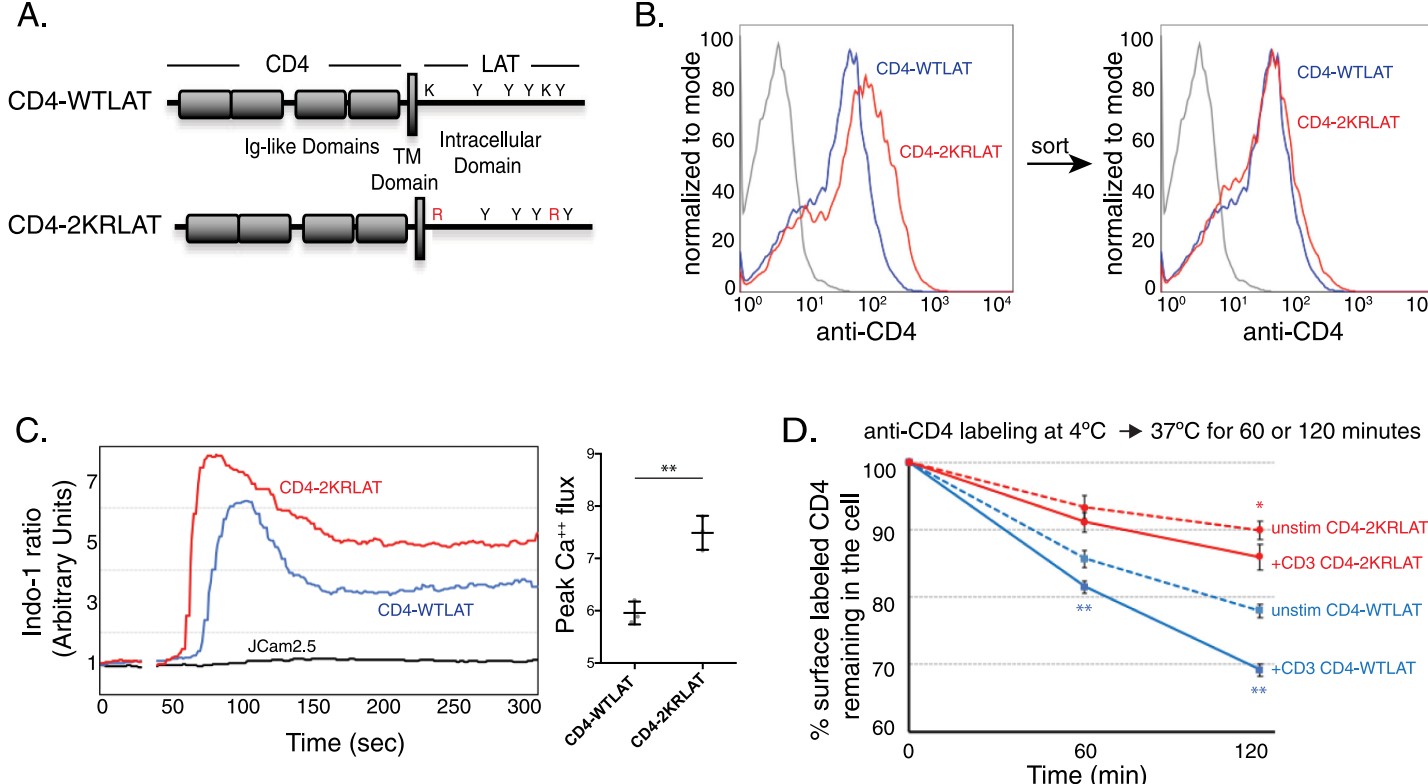

**Fig 1. Characterization of CD4-2KRLAT. A**. Schematic of CD4-WTLAT and CD4-2KRLAT chimeric constructs. **B**. Surface expression of CD4 in JCam2.5 cells stably expressing CD4-LAT and CD4-2KRLAT. **C**. JCam2.5 cells or cells stably transduced with CD4-LAT or CD4-2KRLAT were loaded with Indo-1 AM, stimulated with 10μg /ml OKT3 and assayed for intracellular calcium flux. Left: Representative plots of three independent experiments. Right: Quantitation of peak calcium flux measurements from three independent experiments. Each symbol denotes an individual experiment and bars denote means ± SD for each group. **D**. JCam2.5 cells stably expressing CD4-WTLAT or CD4-2KRLAT were labeled with anti-CD4 (clone OKT4) at 4°C and transferred to 37°C for indicated times at which total CD4 levels were measured by flow cytometry. % surface labeled CD4 remaining in the cell was measured as described in Materials and Methods. Bars denote means ± SEM of three independent experiments. Statistical significance was determine using Student's t test. *p < 0.05; ** p < 0.0005.

of cell surface CD4-WTLAT was TCR dose dependent, while CD4-2KRLAT loss was not significantly affected by TCR dose (S2 Fig).

To disrupt cellular ubiquitination in another way, we used TAK-243, a small-molecule inhibitor of the ubiquitin activating enzyme (UAE), the primary mammalian E1 enzyme that regulates the ubiquitin conjugation cascade [23]. TAK-243 treatment inhibited cellular ubiquitin conjugation in a dose and time dependent manner, with 1μM TAK-243 treatment for 60 minutes being sufficient for depletion of cellular ubiquitin conjugates (S3A Fig). We then treated JCam2.5 cells expressing CD4-WTLAT or CD4-2KRLAT with 1μM TAK-243 for 60 minutes and followed the loss of cell surface LAT using antibody labeling as described for Fig 1D. While TAK-243 treatment did not cause a change in the kinetics of loss of CD4-2KRLAT, loss of CD4-WTLAT was affected. Surface labeled CD4-WTLAT molecules were lost at a significantly lower rate in TAK-243 treated cells and TCR stimulation did not cause an increase in loss of cell surface LAT (S3B and S3C Fig). Thus, in cells treated with the E1 inhibitor, WTLAT behavior resembled that of 2KRLAT. Together, these results suggest that the degradation of cell surface LAT is ubiquitin-dependent. Upon stimulation, LAT ubiquitination levels increase [7] and as a consequence, LAT is degraded at a higher rate. In the absence of ubiquitin moieties on LAT, LAT from the plasma membrane is degraded at a lower rate and degradation levels are independent of T cell stimulation.

## Cell surface LAT traffics to the lysosome in activated T cells

Ubiquitination is a post-translational modification that can act as a sorting signal to deliver proteins to specific cellular destinations [24]. For integral membrane proteins like LAT, ubiquitin attachment can potentially direct their traffic from the cell surface to late endosomes/lysosomes for degradation. As there is a significant increase in degradation of surface WTLAT compared with 2KRLAT (Fig 1D), we next sought to determine whether ubiquitination was a signal for LAT molecules from the cell surface to be trafficked to lysosomes and degraded.

To enable colocalization analysis of LAT molecules with the lysomal marker LAMP-1, we caused accumulation of cargo in the lysosome by inhibiting lysosomal enzymes. Cells stably expressing CD4-WTLAT or CD4-2KRLAT were treated with 1mg/ml leupeptin for 4 hrs. We then incubated leupeptin-treated cells with fluorescently labeled anti-CD4 on ice, to label the pool of LAT located at the plasma membrane. Following labeling and removal of excess antibody with cold buffer washes, cells were resuspended in 37˚C buffer, immediately dropped onto coverslips coated with stimulatory antibody, and fixed after 30 minutes of activation. Leupeptin treatment and labeling of the cells with antibody did not interfere with the activation of cells as demonstrated by the formation of phosphotyrosine positive microclusters (Fig 2A blue label). CD4-WTLAT labeled at the cell surface was recruited and retained in LAMP1-positive lysosomes (Fig 2A upper panel). In contrast, the CD4-2KRLAT mutant was not recruited to LAMP1-positive lysosomes, even though efficient labeling of CD4-2KRLAT was achieved (Fig 2A lower panel). Quantification of the data revealed differences in the colocalization of CD4-WTLAT and CD4-2KRLAT with LAMP1, with CD4-WTLAT displaying significantly higher localization in the lysosomal compartment (Fig 2B). These data indicate that LAT ubiquitination induced by T cell activation is a signal by which LAT from the surface of the cell is efficiently trafficked to lysosomes for degradation.

Next, we evaluated the relative amounts of surface labeled WT and 2KR LAT that were localized in internal compartments upon T cell activation in leupeptin treated cells. As above, cells were treated with 1mg/ml leupeptin for 4 hrs following which surface LAT pools were labeled. Labeled cells were dropped on stimulatory coverslips and cells were imaged after 30 minutes of T cell activation at 37˚C. Clearly, the majority of CD4-WTLAT seemed to be present in internalized vesicles (Fig 2C upper panel, arrowheads), while CD4-2KRLAT appeared to be at the plasma membrane even after 30 minutes of stimulation (Fig 2C lower panel, arrows). Z-stacks through the cell were collected in order to be able to reconstruct the plasma membrane of the entire cell. Reconstruction of the x-z view of cells clearly demonstrated the presence of CD4-WTLAT in internalized vesicles, while CD4-2KRLAT was mostly at the cell surface (Fig 2D). Quantification of fluorescent LAT revealed significantly higher levels of CD4-WTLAT in internal compartments as compared to CD4-2KRLAT (Fig 2E). Thus, in the absence of LAT ubiquitination there is a dramatic increase in surface levels of LAT.

## Endocytosis rates are similar in WT LAT and 2KRLAT, but recycling to the plasma membrane is enhanced in ubiquitin-resistant LAT

Cell surface LAT internalizes, and like all internalized integral membrane proteins, internalized LAT can either recycle back to the plasma membrane or traffic to lysosomes for degradation. However, the relative contribution of these pathways to the maintenance of surface levels of LAT remains to be determined. The significant increase of ubiquitin-defective 2KR LAT on the cell surface (Fig 2D and 2E) could be a consequence of decreased internalization, or increased recycling of internalized molecules to the cell surface. To directly measure endocytosis rates of WT and 2KR LAT, we developed a flow assay that follows the fate of

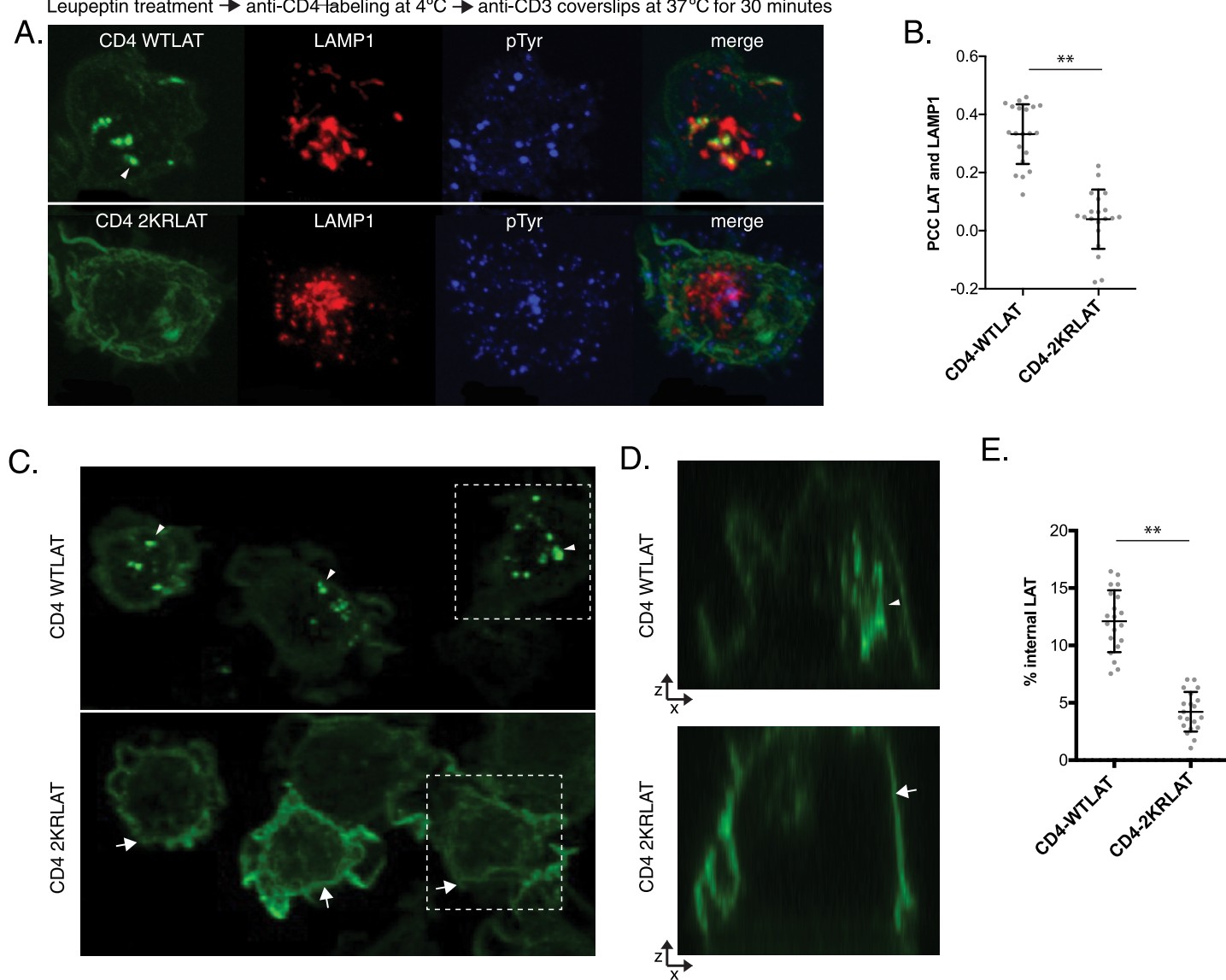

**Fig 2. CD4-WTLAT traffics from the cell surface to the lysosome in activated cells. A**. JCam2.5 cells stably expressing CD4-WTLAT or CD4-2KRLAT were treated with 1mg/ml Leupeptin for 4 hrs. Cells were then labeled with anti-CD4 (clone OKT4) at 4˚C, dropped onto stimulatory coverslips at 37˚C and fixed after 30 minutes. Post-fixation, cells were permeabilized and immunostained for LAMP1 and phosphotyrosine (pTyr). Data is representative of three independent experiments. **B**. Pearson colocalization coefficients between LAT and LAMP1 were quantified. Each symbol denotes an individual cell and bars denote means ± SD for each group. **C**. JCam2.5 cells stably expressing CD4-WTLAT or CD4-2KRLAT were treated with 1mg/ml Leupeptin for 4 hrs. Cells were then labeled with anti-CD4 (clone OKT4) at 4˚C, dropped onto stimulatory coverslips at 37˚C and fixed after 30 minutes. Confocal slices 0.5μm apart were collected through the entire cell. **D**. x-z view of the cells outlined in white in C. Data is representative of three independent experiments. In **A**, **C** and **D** arrowheads indicate internal LAT pools and arrows indicate LAT at the cell surface. **E**. % internal CD4-LAT levels were calculated. Each symbol denotes an individual cell and bars denote means ± SD for each group. Statistical significance was determine using Student's t test. ** p < 0.0001.

surface labeled CD4-WT or CD4-2KRLAT. Cell surface CD4-LAT was labeled using fluorescently tagged anti-CD4 on ice, the cells were washed, and the T cells were re-cultured on ice or at 37˚C. At various indicated times, the remaining surface anti-CD4 was removed by stripping the cell surface antibody using low pH buffer. Flow cytometry revealed the anti-CD4 signal present inside the cell. Quantification of the anti-CD4 signal revealed that the

low pH wash removed 99% of the anti-CD4 antibody of the cells cultured on ice. No significant difference was observed in the rate of endocytosis in CD4-WTLAT and CD4-2KRLAT in either stimulated or unstimulated cells, though endocytosis rates were higher in stimulated cells as has been reported before (Fig 3A; [4, 5]). The same rates of internalization of the 2KR mutant and WTLAT imply that the internalization of LAT does not depend on LAT ubiquitination. Together with our previous studies, our data suggest that though LAT is internalized by a Cbl and ubiquitin-dependent mechanism, ubiquitination of LAT itself is not necessary for LAT internalization.

To follow the recycling of LAT, we once again used our flow cytometry assay. Surface-labeled CD4-WT or 2KRLAT proteins were allowed to internalize for 30 min at 37 ˚C, after which time the cells were returned to ice and the remaining surface anti-CD4 antibodies were removed by treatment with two washes of low pH buffer. This procedure left a cohort of fluorescent molecules inside the cells. Return of internalized molecules to the plasma membrane was assayed by re-culturing the cells at 37 ˚C, and at various times, the cells were either again washed with low pH buffer on ice or left untreated. Flow cytometry was used to the detect the fluorescent anti-CD4 signal, and the difference in signal between untreated cells and low pH buffer stripped cells was used to calculate the amount of LAT that recycled back to the cell surface. Using this assay, we found that internalized 2KRLAT recycles more rapidly than WTLAT. Furthermore, 2KRLAT recycled more than WTLAT over the course of the assay, suggesting that the size of the 2KRLAT recycling pool is increased in the absence of ubiquitination (Fig 3B). Though activation-induced increases in recycling were observed in both WT and 2KRLAT, significant differences were only observed in 2KRLAT cells. Our data demonstrate that 2KRLAT molecules display increased LAT cell surface expression as a result of increased recycling to the plasma membrane instead of being retained in endosomes and/or degraded in the lysosome. This altered trafficking pattern results in a defect in LAT degradation.

Taken together, we suggest the model shown in Fig 3C. Surface WTLAT is constitutively internalized and degraded at a low rate. Upon TCR activation, WTLAT is phosphorylated, ubiquitinated, internalized at higher rates and transported to the lysosome, thus targeting ubiquitinated LAT for degradation. This presumably occurs because surface LAT levels available to potentiate T cell activation need to be tightly regulated. In the absence of LAT lysines in 2KRLAT, LAT is not subject to ubiquitination. It is still internalized at similar rates, but it is not trafficked to the lysosome and degraded. Instead it continues to recycle back to the plasma membrane.

Optimal regulation of LAT surface expression, as is seen with the WTLAT molecule, is crucial to elicit proper immune responses. Minor changes in surface expression levels could have major effects on the outcome of T-cell activation as is seen with the 2KRLAT and in our previously described studies [8–10]. In addition to newly synthesized protein, levels of surface expressed LAT are regulated by endocytosis, recycling, and retrograde trafficking of LAT. Besides previously identified general mechanisms (such as Cbl E3 ubiquitin ligases, ubiquitin and Rab6) that regulate cell surface expression of LAT, we have identified LAT lysine residues as a specific factor for controlling LAT levels at the cell surface. This finding is of potential importance for the development of new therapeutics that will be designed to enhance anti-tumor immunity of T cells, as LAT surface levels have a direct effect on the magnitude of T cell activation. Further, understanding the mechanisms by which LAT is transported to and maintained at the cell surface could lead to the development of novel strategies to increase or decrease its expression and functional effects. In general, an ability to regulate the trafficking pathways of T cell signaling molecules would provide alternate approaches to regulate T cell signaling in immunotherapy.

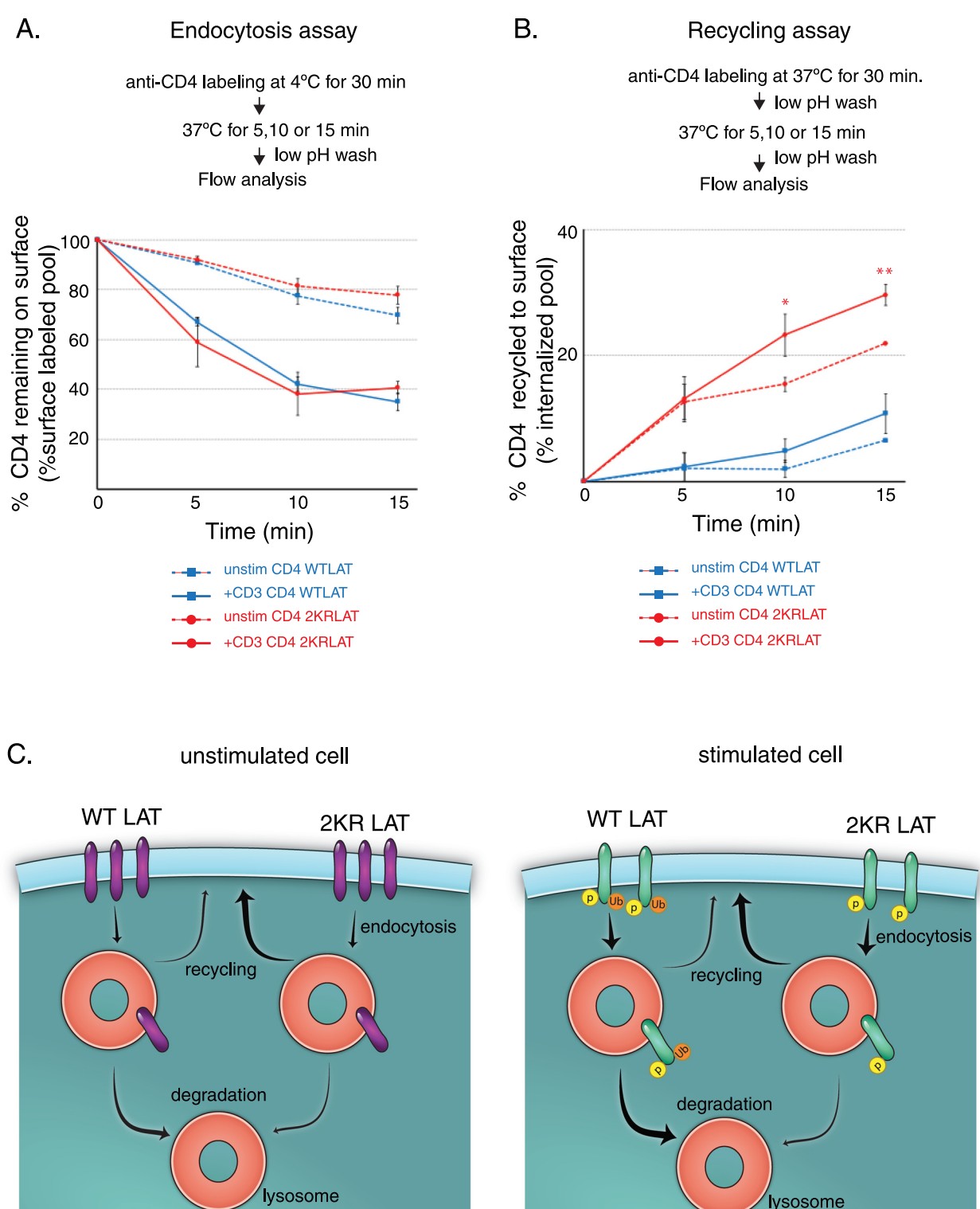

**Fig 3. Endocytosis rates of CD4-WTLAT and CD4-2KRLAT are similar, but 2KR recycles at higher rates. A**. JCam2.5 cells stably expressing CD4-WTLAT or CD4-2KRLAT were labeled with anti-CD4 (clone OKT4) at 4˚C. Cells were transferred to 37˚C for indicated times. Cells were washed with low pH buffer to remove surface CD4-LAT, anti-CD4 levels were measured by flow cytometry and endocytosis rates were calculated as described in Materials and Methods. Data is representative of three independent experiments. **B**. JCam2.5 cells stably expressing CD4-WTLAT or CD4-2KRLAT were labeled with anti-CD4 (clone OKT4) at 37˚C for 30 minutes to generate an internal pool. After a wash with low pH buffer to

remove surface LAT, cells were transferred to 37˚C for indicated times. After a second wash with low pH buffer, fluorescent anti-CD4 levels were measured by flow cytometry and recycling rates were calculated as described in Materials and Methods. Data is representative of three independent experiments. In **A** and **B** bars denote means ± SEM of three independent experiments. Statistical significance was determine using Student's t test. $^*$p < 0.05; $^{**}$p < 0.005. **C**. Model for trafficking of cell surface LAT. In unstimulated cells, cell surface WTLAT is constitutively internalized, recycled and degraded at a low rate (as indicated by the thin arrows). Upon cell stimulation, WTLAT is phosphorylated and ubiquitinated, and compared to unstimulated cells, endocytosed at higher rates, transported to the lysosome at higher rates (as indicated by the thicker arrow), while recycling rates remain the same. As a result, WTLAT gets degraded at a higher rate in stimulated cells. In the absence of LAT lysines, 2KRLAT molecules are endocytosed at similar rates as WTLAT in unstimulated and stimulated cells (as indicated by similar thickness of arrows). However, 2KRLAT is not trafficked efficiently to the lysosome for degradation, whether the cells are stimulated or not. Instead 2KRLAT molecules recycle back to the plasma membrane at higher rates than WTLAT, resulting in higher levels of cell surface 2KRLAT (adapted from [20].

## Supporting information

**S1 Fig. Characterization of CD4-2KRLAT.**
(PDF)

**S2 Fig. Dose-dependent loss of surface LAT.**
(PDF)

**S3 Fig. E1 inhibition of cells expressing CD4-WTLAT or CD4-2KRLAT.**
(PDF)

**S4 Fig. Raw images.**
(PDF)

## Acknowledgments

We thank Karen Wolcott for cell sorting, Paul Kriebel for helpful discussions and Valarie Barr for critically reading the manuscript.

## Author Contributions

**Conceptualization:** Lakshmi Balagopalan.

**Investigation:** Lakshmi Balagopalan, Hiba Malik, Katherine M. McIntire, Joseph A. Garvey, Tiffany Nguyen.

**Methodology:** Lakshmi Balagopalan, Hiba Malik, Katherine M. McIntire, Joseph A. Garvey, Tiffany Nguyen, Ana B. Rodriguez-Peña.

**Project administration:** Hiba Malik.

**Supervision:** Lawrence E. Samelson.

**Validation:** Lakshmi Balagopalan.

**Writing – original draft:** Lakshmi Balagopalan.

**Writing – review & editing:** Lakshmi Balagopalan, Lawrence E. Samelson.

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
