## [Decision Letter · Decision Letter 0]

11 Oct 2019

PONE-D-19-23862

Bypassing ubiquitination enables LAT recycling to the cell surface and enhanced signaling in T cells.

PLOS ONE

Dear Dr. Balagopalan,

Thank you for submitting your manuscript to PLOS ONE. After careful consideration, we feel that it has merit but does not fully meet PLOS ONE’s publication criteria as it currently stands. Therefore, we invite you to submit a revised version of the manuscript that addresses the points raised during the review process.

We would appreciate receiving your revised manuscript by Nov 25 2019 11:59PM. To enhance the reproducibility of your results, we recommend that if applicable you deposit your laboratory protocols in protocols.io, where a protocol can be assigned its own identifier (DOI) such that it can be cited independently in the future. For instructions see: http://journals.plos.org/plosone/s/submission-guidelines#loc-laboratory-protocols

We look forward to receiving your revised manuscript.

Kind regards,

Mariola J Edelmann, Ph.D.

Academic Editor

PLOS ONE

1. Our internal editors have looked over your manuscript and determined that it is within the scope of our Autophagy and Proteostasis Call for Papers. This collection of papers is headed by a team of Guest Editors: Sharon Tooze, Fulvio Regiori and Thorsten Hoope. The Collection will encompass a diverse range of research articles from early initiation of autophagy, to understand the role other proteostasis pathways play in maintaining cellular homeostasis and the cross talk between the two.  Additional information can be found on our announcement page: https://collections.plos.org/s/autophagy-proteostasis..

If you would like your manuscript to be considered for this collection, please let us know in your cover letter and we will ensure that your paper is treated as if you were responding to this call. If you would prefer to remove your manuscript from collection consideration, please specify this in the cover letter.

3.

PLOS ONE now requires that authors provide the original uncropped and unadjusted images underlying all blot or gel results reported in a submission’s figures or Supporting Information files. This policy and the journal’s other requirements for blot/gel reporting and figure preparation are described in detail at https://journals.plos.org/plosone/s/figures#loc-blot-and-gel-reporting-requirements and https://journals.plos.org/plosone/s/figures#loc-preparing-figures-from-image-files. When you submit your revised manuscript, please ensure that your figures adhere fully to these guidelines and provide the original underlying images for all blot or gel data reported in your submission. See the following link for instructions on providing the original image data: https://journals.plos.org/plosone/s/figures#loc-original-images-for-blots-and-gels.

Reviewers' comments:

Reviewer's Responses to Questions

**Comments to the Author**

1. Is the manuscript technically sound, and do the data support the conclusions?

Reviewer #1: Yes

Reviewer #2: Yes

2. Has the statistical analysis been performed appropriately and rigorously? 

Reviewer #1: Yes

Reviewer #2: Yes

3. Have the authors made all data underlying the findings in their manuscript fully available?

Reviewer #1: Yes

Reviewer #2: Yes

4. Is the manuscript presented in an intelligible fashion and written in standard English?

Reviewer #1: Yes

Reviewer #2: Yes

5. Review Comments to the Author

Reviewer #1: The underlying molecular interactions at the initiation of the T cell response has been widely studied. One of the molecules that is implicated in the T cell response is Linker of Activation of T cells (LAT). The mechanism that underlies the turnover of LAT is the focus of the studies in this manuscript. LAT levels at the plasma membrane, its endocytosis after phosphorylation and its recycling back to the cell surface are all points in LAT turnover that can affect the T cell response. There are 2 pools of LAT; at the cell surface and in vesicular compartments. The mobilization of LAT may be regulated by its ubiquitination. In this manuscript, the authors assess the role of ubiquitination on LAT turnover.

To assess ubiquitination, a chimeric molecule was constructed that contained the extracellular domain of CD4 and the cytoplasmic domain of LAT with or without mutated lysines. Following CD4 ligation with anti CD4 antibody or cell activation with anti-CD3, they monitored the redistribution of the chimeric CD4/LAT molecule within the cell. The results of the experiments are clearly presented.

Within the context of these experiments, as an additional evaluation of the ubiquitination of LAT, experiments in the presence of cell permeable ubiquitin inhibitors that target the ubiquitin proteasome system at an early point (an E1 inhibitor) or later in the process (proteasome inhibitor) would be informative. The expectation is that the CD4WTLAT and 2KRLAT will be differentially affected. Those differences should complement the studies reported here.

Reviewer #2: This manuscript needs some revision as there are mistakes /inaccurate statements. Below is what I found. Authors need to proof-read it carefully.

Line 57: Why capitalized Cell

Line 58: Phosphorylation of the TCR on cytosolic tyrosine residues ...... Really ?

Line 61: Wrong reference. The first author seems to love to cite her own papers.

Line 66: leading to activation of the downstream kinases and transcription factors......what kinases down of LAT? why only kinases and transcription factors? Please rewrite this sentence.

Line 67: effector functions as T cell proliferation and cytokine expression. Is T cell proliferation an effector function?

Line 68: wrong reference

Line 81: a defect in protein turnover rate. How a turnover rate could be defective? Should be either decreased or increased!

Line 84: enhance T-cell potency. why we are talking potency here? potent in doing what?

Line 92: move some cited references that reported essential role of surface LAT.

Line 107: LAT endocytosis in these cells. What are these cells? Did you talk about any cells in this paragraph?

Line 110: Molecular switch? a fancy word. perhaps simply targets LAT for degradation.

Line 272: significantly effecting. should be affecting.

Line 276-277: make it clearer for " dose dependent and TCR dose"

Line 277: ubiquitin is a signal for the loss of cell surface LAT. Weird statement

Line 300: labeling of the construct at the cell surface. Construct?

6. PLOS authors have the option to publish the peer review history of their article (what does this mean?). If published, this will include your full peer review and any attached files.

Reviewer #1: No

Reviewer #2: No

---

## [Author Response · Author response to Decision Letter 0]

23 Dec 2019

Our responses to the Academic Editors' requests and Reviewers' comments are included in the "Response to Reviewers" letter.

---

## [Decision Letter · Decision Letter 1]

29 Jan 2020

Bypassing ubiquitination enables LAT recycling to the cell surface and enhanced signaling in T cells.

PONE-D-19-23862R1

Dear Dr. Balagopalan,

We are pleased to inform you that your manuscript has been judged scientifically suitable for publication and will be formally accepted for publication once it complies with all outstanding technical requirements.

With kind regards,

Mariola J Edelmann, Ph.D.

Academic Editor

PLOS ONE

Additional Editor Comments (optional):

Reviewers' comments:

Reviewer's Responses to Questions

**Comments to the Author**

1. If the authors have adequately addressed your comments raised in a previous round of review and you feel that this manuscript is now acceptable for publication, you may indicate that here to bypass the “Comments to the Author” section, enter your conflict of interest statement in the “Confidential to Editor” section, and submit your "Accept" recommendation.

Reviewer #1: All comments have been addressed

Reviewer #2: All comments have been addressed

2. Is the manuscript technically sound, and do the data support the conclusions?

Reviewer #1: Yes

Reviewer #2: Yes

3. Has the statistical analysis been performed appropriately and rigorously? 

Reviewer #1: Yes

Reviewer #2: Yes

4. Have the authors made all data underlying the findings in their manuscript fully available?

Reviewer #1: Yes

Reviewer #2: Yes

5. Is the manuscript presented in an intelligible fashion and written in standard English?

Reviewer #1: Yes

Reviewer #2: Yes

6. Review Comments to the Author

Reviewer #1: (No Response)

Reviewer #2: This study demonstrated that LAT ubiquitination plays an important role in LAT half-life and its function in TCR-mediated signaling. My concerns were all addressed.

7. PLOS authors have the option to publish the peer review history of their article (what does this mean?). If published, this will include your full peer review and any attached files.

Reviewer #1: No

Reviewer #2: No

---

## [Editor Report · Acceptance letter]

13 Feb 2020

PONE-D-19-23862R1 

Bypassing ubiquitination enables LAT recycling to the cell surface and enhanced signaling in T cells. 

Dear Dr. Balagopalan:

I am pleased to inform you that your manuscript has been deemed suitable for publication in PLOS ONE. Congratulations! Your manuscript is now with our production department. 

With kind regards,

on behalf of

Dr Mariola J Edelmann 

Academic Editor

PLOS ONE